# Viral priming of cell intrinsic innate antiviral signaling by the unfolded protein response

Tea Carletti[1,4], Mohammad Khalid Zakaria[1,3,4], Valentina Faoro[1], Laura Reale[1], Yvette Kazungu[1], Danilo Licastro[2] & Alessandro Marcello[1]

The innate response to a pathogen is critical in determining the outcome of the infection. However, the interplay of different cellular responses that are activated following viral infection and their contribution to innate antiviral signalling has not been clearly established. This work shows that flaviviruses, including Dengue, Zika, West Nile and Tick-borne encephalitis viruses, activate the unfolded protein response before transcription of interferon regulatory factor 3 induced genes. Infection in conditions of unfolded protein response priming leads to early activation of innate antiviral responses and cell intrinsic inhibition of viral replication, which is interferon regulatory factor 3 dependent. These results demonstrate that the unfolded protein response is not only a physiological reaction of the cell to viral infection, but also synergizes with pattern recognition sensing to mount a potent antiviral response.

[1] Laboratory of Molecular Virology, International Centre for Genetic Engineering and Biotechnology (ICGEB), Trieste, Italy. [2] CBM scrl, Trieste, Italy. [3] Present address: The Pirbright Institute, Ash Road, Pirbright, Woking, Surrey GU24 0NF, United Kingdom. [4] These authors contributed equally: Tea Carletti, Mohammad Khalid Zakaria Correspondence and requests for materials should be addressed to A.M. (email: marcello@icgeb.org)

Flaviviruses are a family of relevant human pathogens delivered by mosquitoes or ticks. Dengue virus (DENV), Zika virus (ZIKV), West Nile virus (WNV), and tick-borne encephalitis virus (TBEV) are only few examples affecting tropical countries and Europe[1,2].

The genome of Flaviviruses is a single RNA filament of positive polarity encoding a polyprotein precursor, which is processed into structural and nonstructural proteins[3]. The RNA genome is replicated by the viral RNA-dependent RNA polymerase through a complementary template of negative polarity, which forms transient double-stranded RNA (dsRNA) intermediates. Infection induces important rearrangements of cytoplasmic membranes of the endoplasmic reticulum (ER) with the formation of characteristic replication vesicles containing dsRNA and replicative proteins[4–6].

Target cells respond to viral infection by activating innate defense mechanisms. Cytoplasmic pattern recognition receptors (PRRs) recognize viral RNA intermediates as pathogen-associated molecular patterns (PAMPs) to trigger the interferon (IFN)-mediated antiviral response[7]. The innate immune response is generally associated to the activity of IFN leading to the induction of interferon-stimulated genes (ISGs) endowed with antiviral activity. However, at very early time points post infection, cell intrinsic mechanisms of defense play an important role in mediating an antiviral response, while IFN remains essential to protect neighboring uninfected cells and to contain the spread of infection.

Virus infection can also lead to ER stress by unscheduled accumulation of viral proteins or modification of ER membranes. Accumulation of unfolded or misfolded proteins in the ER leads to a stress response by activating the unfolded protein response (UPR) pathway, which restores ER homeostasis[8,9]. Three transmembrane ER proteins mediate the UPR: the inositol-requiring enzyme 1 (IRE1), the protein kinase RNA-like ER kinase (PERK), and the activating transcription factor 6 (ATF6). Activated IRE1 cleaves the X-box binding protein 1 (XBP1) mRNA in the cytoplasm leading to the spliced form, which is translated into an active transcription factor (XBP1s). Activated PERK phosphorylates the eukaryotic translation initiation factor 2α (eIF2α) causing inhibition of protein synthesis, but also enhanced translation of the activating transcription factor 4 (ATF4). ATF4 in turn promotes transcription of several genes including the CCAAT/enhancer-binding protein-homologous protein (CHOP), a proapoptotic transcription factor, and feedback regulators, which counteract the phosphorylation of eIF2α. Activation of the ATF6 pathway leads to the processing of ATF6 into a cleaved product that translocate to the nucleus to activate UPR-regulated genes including Xbp1.

It has been previously shown that sterile co-stimulation of the UPR with certain PAMPs such as lipopolysaccharide or Poly-inosinic:polycytidylic (poly I:C) acid greatly potentiates the expression of IFNβ[10–12]. These observations led to the intriguing hypothesis that UPR and PRR signaling synergize during infection to provide optimal antiviral immunity[13].

In this work, transcriptome analysis of TBEV infected cells shows upregulation of a number of genes involved in the UPR. Careful temporal analysis demonstrates that the IFN response is a late event preceded by the UPR. Most importantly, preactivation of the UPR during flavivirus infection causes a decrease of viral titers, an earlier induction of IRF3 phosphorylation and translocation, and of IFN and ISGs transcription. IRF3 depletion rescues flavivirus replication induced by UPR priming. Furthermore, depletion of IRE1, but not of ATF6 or PERK, enhance viral replication and rescue specifically TBEV from antiviral priming of the UPR. This IRE1 function is independent of its RNAse activity, but dependent on IRF3 and RIG-I. Hence, these data demonstrate that the UPR is a very early cellular response to TBEV infection, which triggers an IRF3/RIG-I dependent cell intrinsic antiviral response through IRE1.

## Results

**Transcriptome analysis of TBEV infection.** In previous studies a consistent delay of the IFN beta (IFNβ) response, following flavivirus infection, was described[14,15]. Viral replication could be detected as early as 8 h post-infection (h.p.i.), while IFNβ mRNA appeared only after 16 h.p.i. Starting from this observation, in order to investigate the cellular pathways that could be activated in the infected cells before IFNβ induction, an unbiased transcriptome analysis of infected cells was conducted. Newly synthesized TBEV RNA was already high as early as 10 h.p.i., a time point, when IFNβ mRNA could not be detected. As expected, IFNβ mRNA was eventually upregulated at 24 h.p.i. These two time points were chosen to conduct the transcriptome analysis of the infected cells. Total RNA was extracted from infected cells in triplicate independent experiments and subjected to high-content sequencing. Differential analysis of the transcriptome of U2OS cells infected with TBEV at 24 versus 10 h.p.i. showed significant upregulation of 437 genes (fold change ≥2) and downregulation of 318 genes (fold change ≤2) with a false discovery rate of <0.05 (DESEQ2 statistical analysis) (Fig. 1a). Ingenuity Pathway Analysis (IPA) indicated the UPR as the most highly significant canonical pathway ($-\log$ ($p$-value) = 7.37) followed by ER stress ($-\log$ ($p$-value) = 3.93). Several genes showed upregulation, including HSPA5 (Heat-Shock 70 kDa Protein 5, BiP), Xbp1/Xbp1s, DDIT3 (CHOP), and the chaperones DnaJ (Hsp40) Homolog, Subfamily C, Member 3 (DNAJC3/P58IPK), and Subfamily B, Member 9 (DNAJB9). To conclude, significant early activation of UPR-related genes was observed during TBEV infection.

**UPR is induced before the interferon response following infection.** The kinetics of UPR induction was analyzed temporally following TBEV infection. As shown in Fig. 1, TBEV infection of U2OS cells was productive as early as 10 h.p.i. (Fig. 1b) with TBEV RNA being detectable at 8 h.p.i. (Fig. 1c). At variance, IFNβ mRNA appears only after 16 h.p.i. (Fig. 1d). UPR genes such as Xbp1, DNAJC3, and DNAJB9 also appear at late time points, concomitantly with IFNβ mRNA (Fig. 1e and Supplementary Fig. 1A and 1B). However, the spliced form of Xbp1 mRNA, an indicator of IRE1 activation, could be detected as early as 12 h.p.i. (Fig. 1f). Similarly, the PERK-dependent activation of CHOP occurred before the IFNβ response (Fig. 1g). Indeed, PERK phosphorylation could be detected at 8 h.p.i. followed by phosphorylation of the eIF2α (Fig. 1h). The third arm of the UPR response is initiated by nuclear translocation of ATF6. To monitor the ATF6 pathway, GFP-ATF6 was transfected in U2OS cells followed by TBEV infection[16]. As shown in Supplementary Fig. 1C and quantified in Fig. 1i, translocation of GFP-ATF6 into the nucleus of infected cells occurred from 8–12 h.p.i. Consistently, UPR genes that are activated downstream of the ATF6 pathway, such as BiP (Fig. 1j), were also induced following TBEV infection.

**Induction of UPR leads to early activation of an innate antiviral response during flavivirus infection.** To summarize the above findings, all three arms of the UPR were activated at early time points following TBEV infection, prior to IFNβ induction. Therefore, the UPR could be a prerequisite for a proper antiviral response. To address this hypothesis, U2OS cells were exposed to Tunicamycin (TM), a well-known inducer of the UPR, immediately after TBEV infection. As shown in Fig. 2a, b, viral yields and

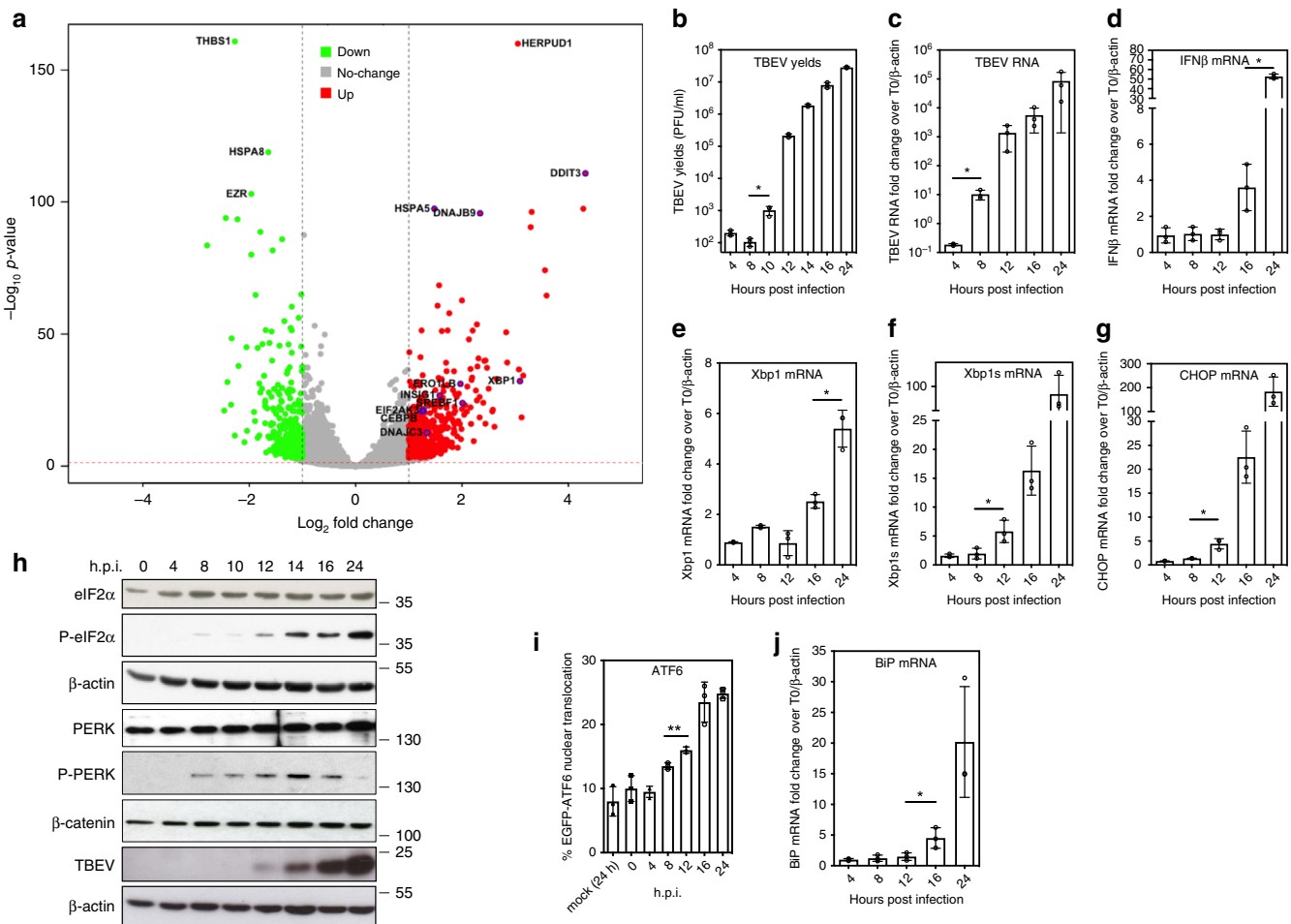

**Fig. 1** Temporal analysis of the UPR response to TBEV infection. **a** Difference in total transcript expression following TBEV infection. Total RNA was extracted from TBEV infected cells at 0, 10, and 24 h.p.i. in triplicate independent experiments and processed for high-content sequencing (~24.6 million reads for each time point). The volcano plot shows the differential gene expression at 24/0 versus 10/0 h.p.i. Horizontal and vertical dashed lines indicate cutoff values (FDR value of 0.05 corresponding to 1,30 Score and absolute logarithmic fold-change >2, respectively). Genes having a significant altered expression are emphasized in red (upregulated) and green (downregulated). Upregulated hits from the UPR pathway are encircled in blue. **b** Time course of viral yields. U2OS cells were infected with TBEV at moi = 1. Supernatant from infected cells were used to infect Vero cells to measure virus yields (PFU/ml). **c–g**, **k** Time course of viral RNA and UPR-related mRNAs. U2OS cells were infected as in **b**. Total RNA extracted at the indicated time points and used as template for qPCR using primers specific for TBEV (5'-NCR) (**c**). TBEV RNA amplification products were normalized to β-actin RNA and plotted as fold change from time 0. Data and statistics are plotted as in Fig. 1b. The same protocol was used to quantify mRNA of IFNβ (**d**), Xbp1 (**e**), Xbp1s (**f**), CHOP (**g**), and BiP (**j**). **h** Time course of PERK activation. U2OS cells were infected with TBEV at moi = 1. At the indicated time points, the total protein content was extracted and subjected to immunoblotting with the indicated antibodies. **j** Time course of ATF6 translocation. Cells treated as in (**b**) were quantified for ATF6 nuclear translocation. GFP positive cells were manually counted for ATF6 nuclear translocation at each time point. The graph shows the results from two independent experiments, 200 cells for each time point. Typically three biologically independent experiments (n = 3) in triplicate repeats were conducted for each condition examined. Average values are shown with standard deviation and p-values, measured with a paired two-tailed t-test. Significant p-values are indicated by asterisks (**p < 0.01; *p < 0.05). Source data are provided as a Source Data file

viral RNA were markedly reduced following TM treatment. Since TM inhibits N-linked glycosylation and could potentially affect viral infectivity, viral RNA levels were also investigated. To note, at 8 h.p.i., while viral yields were not yet increasing (Fig. 2a), there was a significant inhibition of viral replication in the presence of TM (Fig. 2b), demonstrating that this early antiviral effect is independent of any unspecific activity on the glycosylation of viral proteins. As a control of TM activity, Xbp1s mRNA was also induced at early time points in the presence of TM (Fig. 2c). Interestingly, induction of IFNβ mRNA occurred much earlier following preactivation of the UPR. As shown in Fig. 2d, treatment of U2OS cells with TM alone (gray bars) stimulated a weak tenfold increase of IFNβ mRNA only after 24 h of treatment. However, upon both TBEV infection and TM treatment, IFNβ was clearly induced as early as 8 h.p.i. (white bars).

To rule out any unspecific effect of TM, the same approach was repeated using Thapsigargin (TG), which activates UPR by blocking the ER calcium ATPase. As shown in Supplementary Fig. 1D and E, both TM and TG inhibit TBEV, but, while TM showed a partial effect on E protein glycosylation as expected, TG did not affect the viral protein. TG was first verified for being able to induce Xbp1s in U2OS at the concentration used (Supplementary Fig. 1F). Next, cells were infected with TBEV followed by treatment with TG. As shown in Fig. 2e, f, TG behaves similarly to TM in inducing an early IFNβ response and inhibiting virus yields. As observed with TM, IFNβ mRNA was induced at low levels by TG alone after 24 h of treatment, but synergized with TBEV infection to potentiate the innate response.

A similar analysis was conducted on other members of the flavivirus family such as DENV2 (Fig. 2g, h), WNV (Fig. 2i, j),

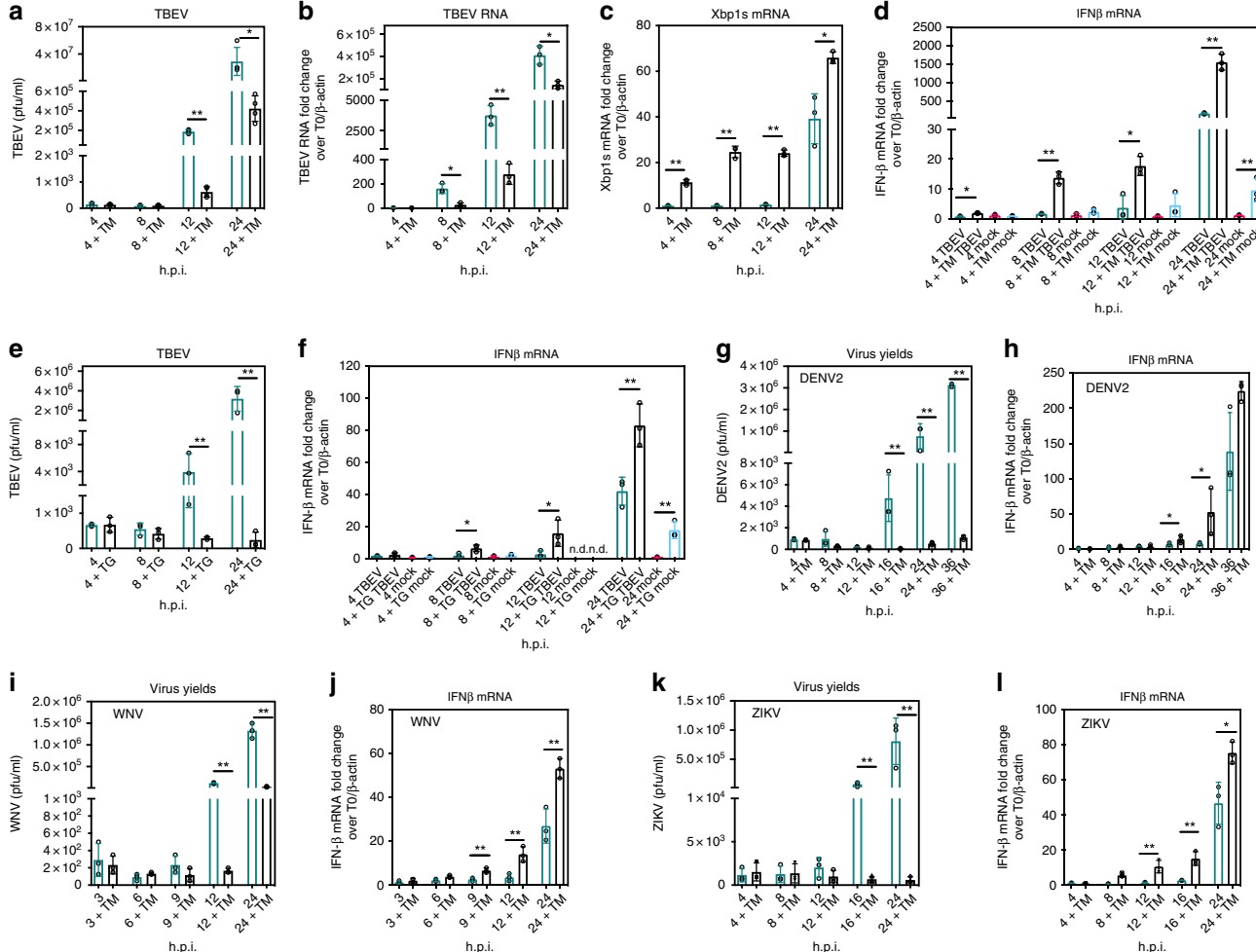

**Fig. 2** Modulation of the interferon response to flavivirus infection by the UPR. **a** Tunicamycin treatment inhibited TBEV yields. U2OS cells were either infected with TBEV at moi = 1 (blue bars) or treated with 1 µg/ml Tunicamycin (TM) immediately after infection (black bars). At the indicated time points, supernatants from infected cells were used to infect Vero cells to measure virus yields (PFU/ml). **b–d** Tunicamycin treatment inhibited TBEV RNA, increased Xbp1s and led to early IFNβ induction. U2OS cells were infected as in **a**. Total RNA was extracted at the indicated time points and used as template for qPCR using primers specific for TBEV 5'-NCR (**b**), Xbp1s (**c**) or IFNβ (**d**). β-actin was used for normalization and data plotted as fold change from time 0. Values of IFNβ mRNA from mock-infected cells treated with Tunicamycin alone were also indicated (**d**, azure bars). **e, f** Thapsigargin treatment inhibited TBEV yields and led to early IFNβ induction. U2OS cells were infected with TBEV at moi = 1 (blue bars) or treated with 0.5 µM Thapsigargin (TG) immediately after infection (black bars) and treated as above (**b, d**). **g–l** Tunicamycin treatment inhibited DENV2, WNV, and ZIKV yields and led to early IFNβ induction. U2OS cells were either infected (blue bars) or treated with Tunicamycin (TM) immediately after infection (black bars). At the indicated time points, supernatants from infected cells were either used to infect Vero cells to measure virus yields (DENV2 **g**, WNV **i**, ZIKV **k**) or total RNA was extracted and used as template for real-time qPCR to quantify IFNβ mRNA (DENV2, **h**; WNV, **j**; ZIKV, **l**). Statistics as already described in the legend of Fig. 1. Source data are provided as a Source Data file

and ZIKV (Fig. 2k, l). Likewise, induction of the UPR by TM resulted in a decrease of virus yields and an early IFNβ response demonstrating that early activation of the UPR leading to a sustained IFNβ response is a general mechanism within flaviviruses.

**The UPR-induced antiviral response is independent of canonical interferon signaling.** Since early IFN activation was consistently observed following flavivirus infection in the presence of UPR inducers, it was plausible to hypothesize that the decrease of virus titers could depend on interferon-dependent antiviral signaling. In order to address this, primary embryonic murine fibroblasts, derived from IFN receptor knockout mice (MEF *ifnar-/-*), were infected in the same experimental conditions. Noteworthy, as shown in Fig. 3, wild-type MEF reproduced the same phenotype observed as in U2OS cells, reinforcing the

previous observations also in a nontransformed primary murine cell model. MEF *ifnar-/-* showed an antiviral response and an early induction of IFNβ in the presence of TM, which was kinetically similar to that observed in wild-type MEFs (Fig. 3a, b). This observation points to an IFN-signaling independent antiviral activity.

A subset of ISGs, with antiviral activity, that are known to be induced directly by the transcription factor IRF3, have been previously described[17,18]. IRF3 phosphorylation, measured following infection, occurred at earlier time points and to a greater magnitude in TM-treated cells as compared with control (Fig. 3c). Consistently, nuclear translocation of IRF3 was significantly increased at 8 h.p.i. compared with infection alone in the same experimental conditions (Fig. 3d, e). Transcriptional induction of IRF3-dependent ISGs, such as IFIT1 or Viperin, showed early activation kinetics in the presence of TM, comparable with what was observed for IFNβ, (Fig. 3f, g).

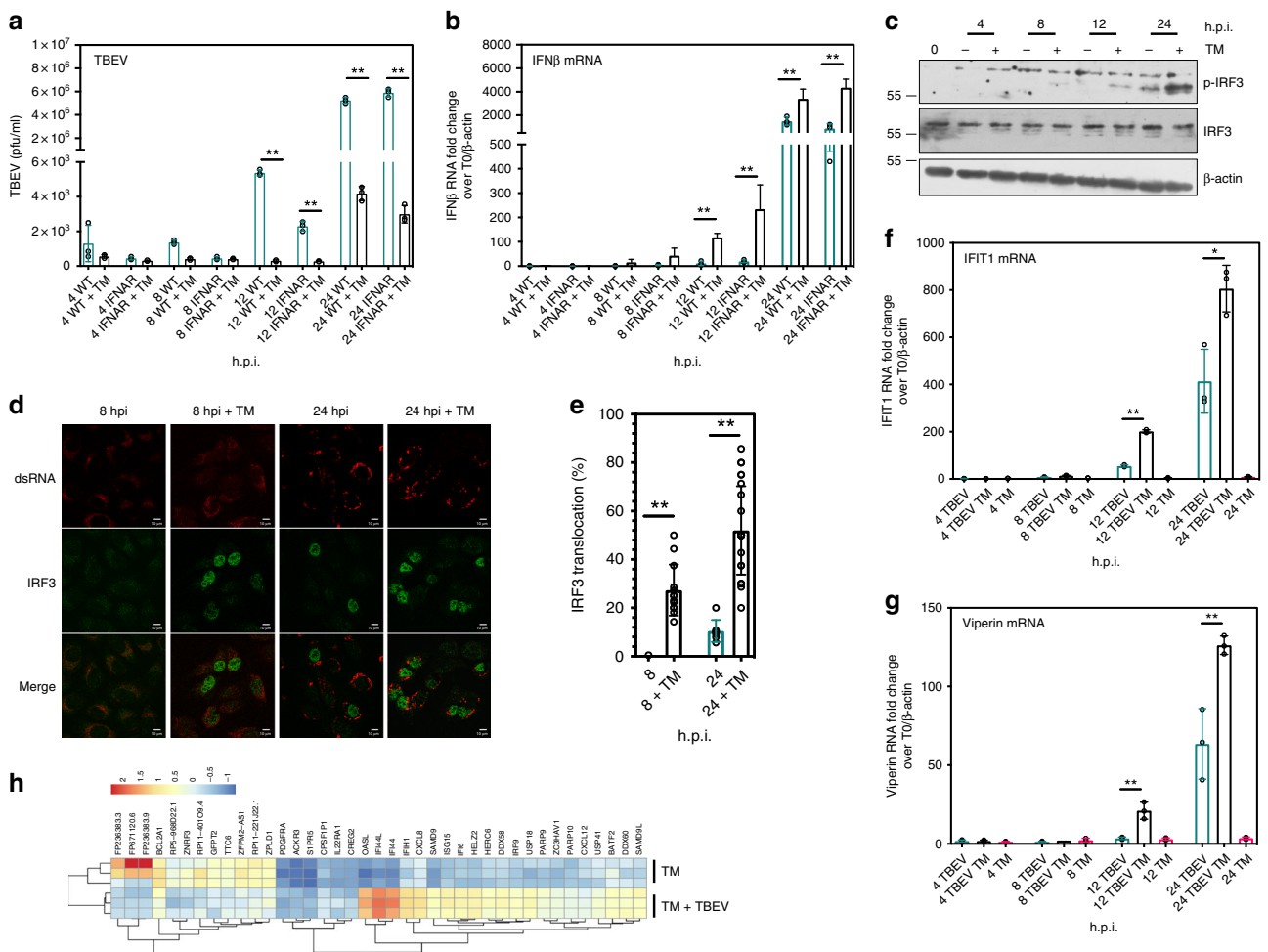

**Fig. 3** UPR-induced antiviral signaling following infection is independent of interferon signaling. **a** Tunicamycin treatment inhibited TBEV yields independently of IFNAR. Primary embryonic MEF *Ifnar⁻/⁻* cells were either infected with TBEV (blue bars) or treated with TM immediately after infection (black bars). At the indicated time points virus yields were measured (PFU/ml). **b** Tunicamycin treatment of TBEV infected cells led to early IFNβ induction. Total RNA from MEF *Ifnar⁻/⁻* infected as above in **a** was extracted at the indicated time points and used as template for real-time qPCR using primers specific for IFNβ. **c** Tunicamycin treatment led to early IRF3 phosphorylation. U2OS cells were infected with TBEV and treated with TM immediately after infection. At the indicated time points the total protein content was extracted and subjected to immunoblotting with the indicated antibodies. **d** Tunicamycin treatment led to early IRF3 nuclear translocation. U2OS cells were infected with TBEV and treated with TM immediately after infection. At the indicated time points, cells were fixed and stained for IRF3 (AlexaFluor 488, green) and dsRNA (AlexaFluor 594, red). Scale bar = 10 μm. **e** Quantification of IRF3 translocation following TBEV infection and Tunicamycin treatment. Cells treated as in **d** were manually quantified for IRF3 nuclear translocation at each time point. The graph shows the results from two independent experiments, ~ 200 cells for each time point. **f**, **g** Tunicamycin treatment of TBEV infected cells led to early ISG induction. Total RNA from U2OS cells infected with TBEV moi = 1 was extracted at the indicated time points and used as template for real-time qPCR using primers specific for IFIT1 or Viperin. Values of ISG mRNA from mock-infected cells treated with Tunicamycin were also indicated (magenta bars). **h** Tunicamycin treatment of TBEV infected cells led an early ISG signature. Transcriptome analysis of cells infected with TBEV compared with infected and treated with TM was performed at 8 h.p.i. in U2OS cells. The cluster of 39 genes differentially regulated in the two conditions is shown in the heatmap (upregulated, red, downregulated, blue, following the indicated gradient). Statistics as already described in the legend to Fig. 1. Source data are provided as a Source Data file

In order to better characterize this phenomenon a transcriptome analysis was conducted on: (i) cells infected with TBEV at 8 h.p.i.; (ii) cells treated with TM for 8 h and (iii) cells both infected and treated with TM at the same time point. IPA indicated the UPR as the most highly enriched canonical pathway (*p*-value = $7.97^{-12}$) induced by sterile TM treatment, while for TBEV infected cells alone at 8 h.p.i. no enriched pathways were indicated, which is in line with the observation that UPR is activated from 10 h.p.i. (Fig. 1). IPA indicated 'activation of IRF by cytosolic PRR" as the most highly enriched canonical pathway (*p*-value = $3.04^{-05}$) for the dataset of cells infected with TBEV and treated with TM. Other top-ranked pathways were 'interferon signaling", 'role of PRRs in recognition of bacteriae and

viruses", and 'role of RIG-I like receptors in antiviral innate immunity" strongly supporting the hypothesis. Differential analysis of the two groups, i.e., treated with TM alone and infected +TM, indicated that the 39 genes that were differentially modulated in the two conditions partition in two functional clusters that defined each condition as clearly visible in the heatmap (Fig. 3h). The cluster of genes differentially regulated during infection in conditions of UPR induction showed a high prevalence of ISGs. In particular ISG15, IFI6, IFI44L, IFI44, IFIH1, DDX60, SAMD9, SAMD9L, ZC3HAV1, PARP10, DDX58, and OASL have been already described as antiviral effectors[19–23]. A subset of these ISGs that include OASL, IFIH1, IFI44, and IFI44L have been validated by RT-qPCR

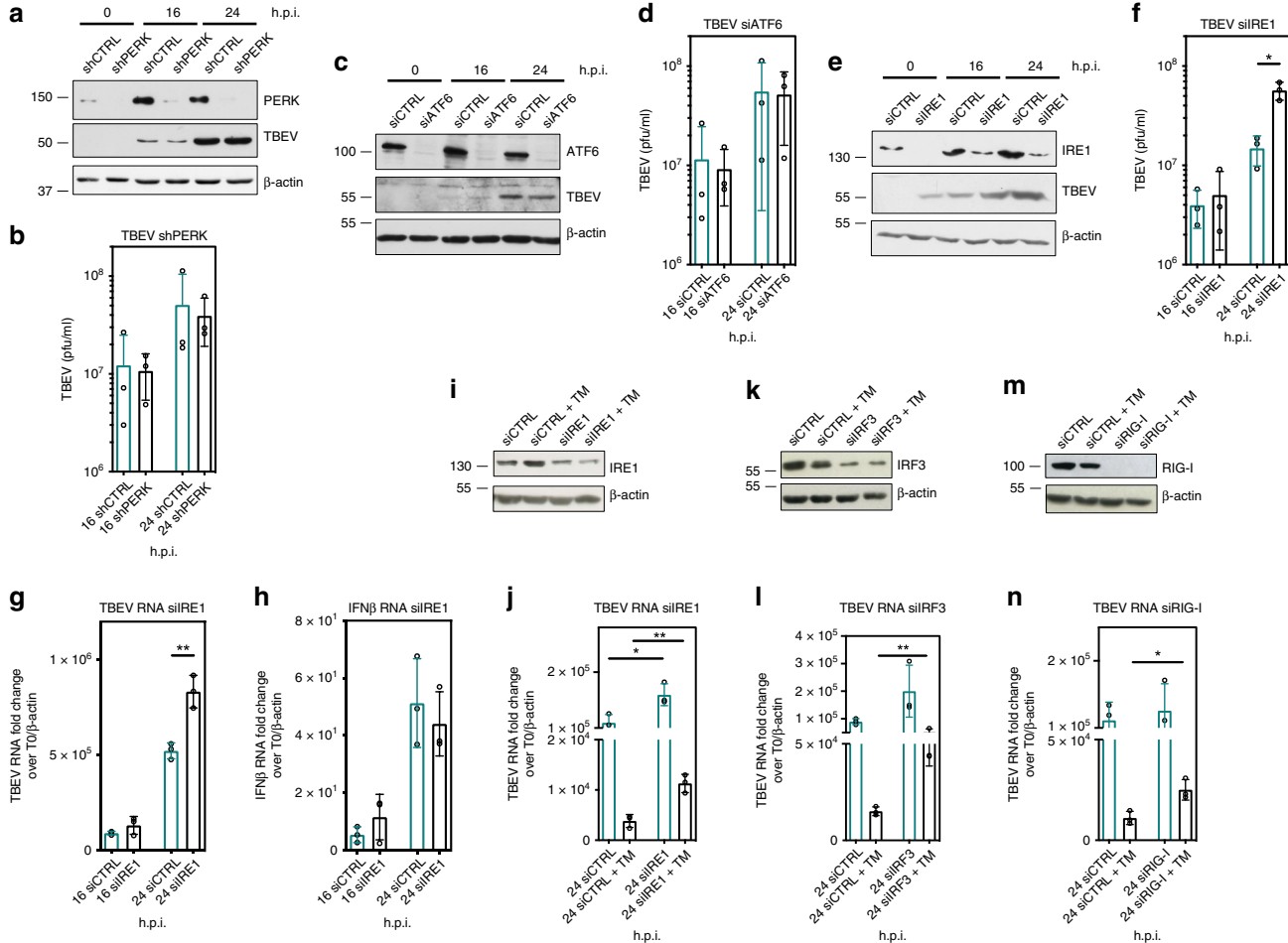

**Fig. 4** UPR-induced antiviral signaling following TBEV infection requires the IRE1 pathway. **a, b** PERK depletion did not affect TBEV yields. U2OS cells stably transduced with shPERK under puromycin selection were infected with TBEV and harvested for immunoblotting at 16 and 24 h p.i., Actin is the protein loading control (**a**). At the indicated time points, supernatants from infected cells were used to infect Vero cells to measure virus yields (**b**). **c, d** ATF6 depletion did not affect TBEV yields. U2OS cells were transfected with siATF6/siCTRL and after 48 h infected with TBEV. Cells were harvested for immunoblotting at 16 and 24 h.p.i. Actin is the protein loading control (**c**). At the indicated time points, supernatants from infected cells were used to infect Vero cells to measure virus yields (**d**). **e, f** IRE1 depletion increased TBEV yields. U2OS cells were treated as above (**c, d**) with siIRE1. **g, h** IRE1 depletion increased TBEV replication but did not affect IFNβ mRNA. U2OS cells were depleted for IRE1 as above (**e**). Total RNA was extracted at the indicated time points and used as template for real-time qPCR using primers specific for TBEV (**g**) or IFNβ (**h**). TBEV RNA amplicons were normalized to β-actin RNA and plotted as fold change from time 0. **i, j** IRE1 depletion partially rescued TBEV replication in TM-treated cells. U2OS cells were either infected with TBEV (blue bars) or treated with TM immediately after infection (black bars) in conditions of IRE1 depletion. Immunoblot for IRE1 and TBEV RNA quantification was performed at 24 h.p.i. **k, l** IRF3 depletion partially rescued TBEV replication in TM-treated cells. U2OS cells were treated as above (**i, j**) in conditions of IRF3 depletion. Immunoblot for IRF3 and TBEV RNA quantification was performed at 24 h.p.i. **m, n** RIG-I depletion partially rescued TBEV replication in TM-treated cells. U2OS cells were treated as above (**i, j**) in conditions of RIG-I depletion. Immunoblot for RIG-I and TBEV RNA quantification was performed at 24 h.p.i. Statistics as already described in the legend to Fig. 1. Source data are provided as a Source Data file

(Supplementary Fig. 2). Therefore, an ISG antiviral signature characterized the early response triggered by the UPR during TBEV infection.

**The UPR-dependent anti-TBEV response is IRE1 and IRF3/RIG-I dependent.** The UPR activates three signaling pathways mediated by the ER transmembrane proteins ATF6, IRE1, and PERK. Each of them was targeted by RNAi to investigate their involvement in antiviral signaling following TBEV infection. U2OS cells were treated with siRNA (IRE1 and ATF6) or with shRNA (PERK) to deplete cells of the respective proteins. As shown in Fig. 4a–d, depletion of PERK or ATF6 did not result in significant changes of viral yields. Depletion of PERK was also obtained by siRNA transfection with similar results (Supplementary Fig. 3A and 3B). Conversely, depletion of IRE1 resulted in a higher TBEV titer (Fig. 4e, f) and enhanced TBEV RNA

levels (Fig. 4g), but not IFNβ mRNA (Fig. 4h). Therefore, the IRE1 pathway appears principally responsible for UPR-mediated antiviral signaling for TBEV. Indeed, IRE1 depletion partially rescued replication of TBEV following TM treatment (Fig. 4i, j). From previous studies it was discovered that the IFN response against TBEV infection depends on the activity of the PRR RIG-I leading to IRF3 induction[15]. In order to understand if the antiviral activity triggered through IRE1 was dependent on IRF3, U2OS cells were treated with a siRNA against IRF3 and then infected and subjected to TM treatment. As shown in Fig. 4k, l, IRF3 depletion almost completely rescued viral replication in conditions of UPR priming. Hence, the UPR primes infected cells for IRF3-mediated antiviral activity. Depletion of RIG-I also partially rescued TBEV replication indicating that PRR sensing contributes to this pathway together with IRE1 (Fig. 4m, n). One possible mechanism could be related to the RNAse activity of

IRE1, which leads to Xbp1 splicing and IRE1-dependent decay of RNA (RIDD) activity. Xbp1 transcription factor has been shown to directly activate IFN, while the RIDD activity has been proposed to generate substrates for RIG-I signaling[24,25]. However, pretreatment of TBEV infected cells with 4μ8C, a specific inhibitor of IRE1 RNAse activity capable of inhibiting Xbp1 splicing (Supplementary Fig. 3C), didn't show any modulation of virus infectivity (Supplementary Fig. 3D). These data indicate that IRE1-mediated pathways other than Xbp1 splicing and RIDD activity need to be considered to explain the UPR-dependent antiviral signaling induced by TBEV infection.

Other flaviviruses, such as WNV, DENV2, and ZIKV, which have been shown previously to be inhibited by early induction of the UPR with TM (Fig. 2g–l), also respond in terms of rescued viral replication in conditions of IRF3 depletion (Supplementary Fig. 4A–4F). These experiments reinforce the notion that there is a causal link between virus-induced ER stress and innate immune sensing and that this feature is shared among different flaviviruses. However, WNV replication inhibited by TM could not be rescued in IRE1 depleted cells (Supplementary Fig. 4G and 4H). This observation indicates that while TBEV appears to depend principally on the IRE1 pathway, other flaviviruses may require the contribution of ATF6 or PERK.

## Discussion

In this work, virus-induced UPR is shown to play a pivotal role in the cell intrinsic innate antiviral response to flaviviruses. UPR and innate sensing have been shown previously to synergize following sterile stimulation[10–13]. However, evidence of this mechanism in infected cells is lacking. The experimental evidence presented here in the context of flavivirus infection points to a direct role of the UPR to trigger a suboptimal activation of the IRF3 pathway, which synergizes with PRR signaling to mount a potent antiviral defense.

This work stems from previous observations that identified a delayed IFN response following flavivirus infection[14,15]. Transcriptome analysis of TBEV infected cells identified the UPR and ER stress pathways as early responses of the cell to infection prior to the IFN response. Similar findings were recently observed for cells of neuronal origin infected with TBEV[26]. Hence, a temporal analysis of UPR activation was performed to better understand the order of cellular events that follow infection.

ATF6 nuclear translocation, following TBEV infection, increased from 8 h.p.i. (Fig. 1i) confirming earlier observations taken at 24 h.p.i.[27]. ATF6-dependent genes, such as Xbp1 and BiP, also increased following infection (Fig. 1e, j). However, ATF6 depletion did not affect viral yields indicating that the ATF6 pathway is dispensable during infection (Fig. 4c, d). These observations are consistent with DENV2 infection, which has been shown to induce ATF6 nuclear translocation but was not affected in ATF6 MEF knockouts[28,29]. Conversely, WNV$_{KUN}$ infection, which also induces activation of the ATF6 branch of the UPR at 12–18 h.p.i., showed decreased titers in ATF6$^{-/-}$ MEFs[30,31]. ATF6 could have a cytoprotective role during milder infections such as WNV$_{KUN}$, while remains nonessential for more pathogenic viruses such as DENV or TBEV. Indeed, a more lethal strain of WNV has been shown to degrade ATF6 in a proteasome-dependent manner[32].

PERK phosphorylation, following TBEV infection, was visible from 8 h.p.i. and eIF2α phosphorylation increased from 12 h.p.i. (Fig. 1h). PERK depletion did not affect viral yields (Fig. 4a, b). Conversely, WNV$_{KUN}$ infection did not induce strong phosphorylation of eIF2α, but infection of PERK KO cells led to an increase of viral replication[30]. Similarly, PERK was shown to negatively regulate DENV2 infection and induced

phosphorylation of eIF2α at 9 h.p.i., but at later time the phosphorylation of this factor was negatively regulated[28]. Another report suggested a proviral role of PERK in DENV2 infection with decreased virus titers in PERK$^{-/-}$ MEFs[33]. These data suggest that TBEV, WNV, and DENV2 are regulating the PERK pathway during infection, but its role is still a matter of debate and may depend upon the cell type used or to the virus with differential kinetic properties.

Xbp1 splicing during TBEV infection was activated as early as 12 h.p.i. (Fig. 1f). Earlier studies already demonstrated Xbp1 splicing following TBEV infection, but only at later time points[27]. Other Flaviviruses, such as WNV$_{KUN}$, DENV2, and JEV, also activate Xbp1 splicing early during infection[28–30]. The transcription activity of Xbp1s is preserved in TBEV infected cells, as demonstrated by the activation of its target genes DNAJB9 and DNAJC3 (Supplementary Fig. 1A and 1B). This is in agreement with what has been observed for JEV and DENV2[29]. Notably, IRE1 depletion resulted in an increase of TBEV titers suggesting a role in antiviral signaling (Fig. 4e, f). These data are in contrast to DENV infection that has been shown to yield significant lower infectious virus in IRE1$^{-/-}$ MEFs[28]. However, besides being different viruses that adapted differently to the host, genetic knockout may activate compensatory pathways that influence the outcome of the assay. Interestingly, several reports indicate that targeting the Xbp1 pathway downstream of IRE1 had no effect on the infection by DENV, JEV, and WNV$_{KUN}$[28,30,32]. Also for TBEV, the use of 4μ8C, a specific inhibitor of the RNAse activity of IRE1 required for Xbp1 splicing, did not impact infectivity suggesting alternative explanations (Supplementary Fig. 3C and 3D).

Next, the causal link between the UPR and innate responses was explored. UPR induction in the context of TBEV infection led to phosphorylation of IRF3 and its nuclear translocation at early time points (Fig. 3c–e). Transcriptome analysis following viral infection and UPR induction caused an early signature of innate response with the activation of several ISGs with antiviral activity (see heatmap in Fig. 3h and Supplementary Fig. 2). These observations point to a direct role of the UPR, in particular through the IRE1 arm for TBEV, to trigger a suboptimal activation of the IFNβ pathway, which then synergizes with PRR signaling to mount antiviral defenses. Indeed, depletion of IRE1 or IRF3/RIG-I in the context of UPR induction preserved from the antiviral effect (Fig. 4). IRF3 depletion induced a rescue of viral replication also for WNV, DENV2, and ZIKV, indicating a general mechanism of UPR priming of antiviral innate immunity. However, IRE1 dependence could be demonstrate for TBEV, but not for WNV (Supplementary Fig. 4G and 4H). This observation is in line with the differences in UPR response among flaviviruses as mentioned above and will require further analysis.

PRR signaling is believed to depend on the unmasking of specific viral PAMPs during infection, particularly RNA replication intermediates for flaviviruses[15]. Disruption of these compartments that allow PRR access to agonist RNA is therefore required for full activation of the IFN response. It is possible to conceive that the UPR also triggers the modification of membranes or membrane-associated protein complexes that eventually unmask the viral RNAs allowing PRR function. Recent loss-of-function screens identified novel ER-associated factors required for flavivirus replication, which could provide hints on the cellular factors involved in this process[34,35].

To conclude, several lines of evidence point to a direct relationship between the UPR and PRR-mediated activation of cell intrinsic innate antiviral signaling. Independent pathways cooperate to respond to viral infection and to overcome viral subversion strategies in the incessant battle between flaviviruses and the host cell.

## Materials and methods

**Cells and viruses**. TBEV represents a good model of the Flavivirus genus because it is easily manipulated leading to robust infection in vitro. Furthermore, U2OS cells were shown previously to maintain an intact PRR-IFNβ signaling pathway with respect to TBEV infection[15].

Vero clone E6 (ATCC C1008) and human osteosarcoma U2OS cells (ATCC HTB-96) were grown under standard conditions in Dulbecco's modified Eagle medium (DMEM) supplemented with 10% fetal bovine serum. Primary murine embryonic fibroblasts (MEF *Ifnar-/-*) from mice lacking the α-chain of the IFN-α/β receptor on a B6 background were kindly provided by U. Kalinke (TWINCORE Germany). Pregnant female mice at 13–14 days of gestation were sacrificed and uterus were removed with the help of forceps and washed with sterile PBS to remove blood. Embryos were carefully separated from the uterus of pregnant mice at 13–14 days of gestation and their head and liver was removed. Each embryo was minced, treated with trypsin EDTA and plated in growth medium. Low-passage MEFs were used for the experiments. Wild-type MEF were similarly obtained from isogenic B6 animals. Animal care and treatments were conducted in conformity with institutional guidelines in compliance with national and international laws and policies.

Working stocks of TBEV (strain Neudoerfl), West Nile virus (strain NY99), Dengue virus type 2 (strain New Guinea), and Zika virus (strain Uganda #976) were routinely propagated and titrated on Vero E6 cells.

UPR inducers Tunicamycin or Thapsigargin (Sigma-Aldrich) were added to cells 1 h after infection (0 h.p.i.). IRE1 Inhibitor 4μ8C (Tocris) was added to U2OS at 0 h.p.i. at a concentration of 30 μM.

**LV Production and shRNA delivery**. Lentiviral silencing vectors were derived from pLKO.1 TRC (Addgene). The control short hairpin RNA (shCTRL) was the pLKO.1 scramble from Addgene. For PERK targeting, a specific sequence was designed and cloned into pLKO.1 TRC (shPERK) using the oligonucleotides 5′-CC GGGGAACGACCTGAAGCTATAAACTGCAGTTTATAGCTTCAGGTCGTTC CTTTTTG-3′ and 5′-AATTCAAAAAGGAACGACCTGAAGCTATAAACTGCA GTTTATAGCTTCAGGTCGTTCC-3′.

Packaging in HEK 293T cells was performed according to standard procedures using the packaging plasmid psPAX2 and pMD2.G (Addgene). Cell supernatants were filtered and kept at −80 °C until use. U2OS cells were transduced in presence of polybrene (hexadimethrine bromide, Sigma-Aldrich) following manufacturer's protocol. Transduced cells were selected using 2 μg/ml puromycin.

**RNA interference**. Pools of small interfering RNAs (siRNAs) were obtained from Dharmacon. ON-TARGET plus Nontargeting Pool was used as a control in all experiments. ON-TARGET plus SMARTpool siRNA was used for the depletion of ATF6 IRE1, PERK, IRF3, and RIG-I. U2OS cells were transfected with siRNAs at a concentration of 100 nM, using Lipofectamine RNAiMAX transfection Reagent (Invitrogen) according to the manufacturer's instructions.

**Western-blotting and immunofluorescence**. For immunoblotting, whole-cell lysates were resolved by SDS-PAGE and blotted onto nitrocellulose membranes. The membranes were blocked in 4% nonfat milk in Tris-buffered saline (TBS) plus 0.1% Tween 20 (TBST), followed by incubation with the primary antibody diluted in the same solution at 4 °C overnight. After washing three times with TBST, secondary horseradish peroxidase (HRP)-conjugated antibodies were incubated for 1 h at room temperature. The blots were developed using a chemiluminescent HRP substrate (Millipore). For immunofluorescence (IF), cells were washed with PBS, fixed with 4% paraformaldehyde for 15 min, incubated for 5 min with 100 mM glycine, and permeabilized with 0.1% Triton X-100 for 5 min. Subsequently, the cells were incubated at 37 °C for 30 min with PBS, 1% bovine serum albumin, and 0.1% Tween 20 before incubation with antibodies. The coverslips were rinsed three times with PBS, 0.1% Tween 20 (washing solution) and incubated for 1 h with secondary antibodies. The coverslips were finally washed three times with washing solution and mounted on slides using Vectashield mounting medium (Vector Laboratories). Fluorescence images of fixed cells were captured on a Zeiss LSM510 Meta confocal microscope with a 63 × numerical-aperture 1.4 Plan-Apochrom oil objective. Further details can be found in previous publications[36,37]. The following antibodies were used in this study: a rabbit polyclonal against the TBEV E protein produced in our laboratory following immunization with the inactivated virions (1:100 IF, 1:1000 WB); a rabbit polyclonal against TBEV prM kindly provided by Franz Heinz, Vienna (1:100 WB); a rabbit polyclonal against human eIF2α from SCBT (1:100 WB, cat.no. sc-11386); a rabbit polyclonal against phosphorylated (Ser51) human eIF2α from Cell Signaling (1:500 WB, cat.no. 9721); a rabbit polyclonal against human PERK from SCBT (1:500 WB, cat.no. sc-13073); a rabbit monoclonal against human PERK from Cell Signaling (1:1000 WB, cat.no. 3192); a rabbit polyclonal against phosphorylated (T981) human PERK from SCBT (1:200 WB, cat.no. sc-32577); a mouse monoclonal against human PKR from SCBT (1:200 WB, cat.no. sc-6282); a mouse monoclonal against human β-catenin from BD Transduction Lab (1:2000 WB, cat.no. 610153); a mouse monoclonal against human β-actin conjugated with peroxidase from Sigma-Aldrich (1:50000 WB, A3854); a rabbit monoclonal against IRF3 from Cell Signaling (1:1000 WB, cat.no. 4302); a mouse monoclonal against dsRNA from English & Scientific Consulting

(1:100 IF, cat.no. J2-1101) a rabbit monoclonal against phospho-IRF3 from Cell Signaling (1:500 WB, cat.no. 4947); a mouse monoclonal against ATF6 from Abcam (1:500 WB, cat.no. ab122897); a rabbit monoclonal against IRE1 from Cell Signaling (1:1000 WB, cat.no. 3294); a mouse monoclonal against RIG-I from Adipogen (WB 1:500, cat. No. AG-20B-0009). Secondary antibodies conjugated with AlexaFluor 488/594 were from Life Technologies (1:500 IF, cat.no. anti-mouse 594 A21207 and anti-rabbit 488 A-21206) and peroxidase conjugates from Dako (1:5000 WB, cat.no. anti-rabbit P0448 and anti-mouse P0447).

**Real-time quantitative reverse transcription PCR**. For real-time quantitative reverse transcription PCR (qPCR) total cellular RNA was extracted with the UPzol according to the manufacturer's protocol (Biotechrabbit) and treated with DNase I (Invitrogen). 500 ng were then reverse-transcribed with random primers and M-MLV Reverse Transcriptase (Invitrogen). Quantification of mRNA was obtained by real-time PCR using the Kapa Sybr fast qPCR kit on a CFX96 Bio-Rad thermo-cycler. Primers for amplification are listed in Supplementary Table 1.

**Transcriptome analysis by RNAseq**. For the first transcriptome analysis (Fig. 1) Human U2OS cells were infected with TBEV at a moi of 1 PFU/cell. Cells were lysed using UPzol (Biotechrabbit) and total RNA was extracted at time 0, 10, and 24 h post-infection (h.p.i.). For the second transcriptome analysis (Fig. 3) U2OS cells were infected with TBEV at a moi of 1 PFU/cell and after 1 h media were replaced with normal growth medium or DMEM plus TM. Cells were lysed with UPzol and total RNA was extracted from each condition at 0 and 8 h.p.i.

Quality of extracted RNA was checked by gel electrophoresis (ribosomal 18S and 28S), spectrophotometric analysis (260/280 > 1.8), and Agilent bioanalyzer (RNA integrity number, RIN ≥ 8). All cDNA libraries of polyA-containing mRNA molecules were prepared using Illumina TruSeq standard protocol. Libraries were pooled and sequenced on two different Illumina Platforms. The first run was performed on Hiseq2000 4-plex run single reads, 50 bp reads, while the second run was performed on HiscanSQ 8-plex run pair-end reads, 2 × 100 bp reads. All data were subjected to quality control using FastQC software. Single reads were mapped against the human genome RNA reference from NCBI using CLCbio software, while pair-end reads were mapped against Homo sapiens GRCh38.77 reference from UCSC using STAR software[38]. Bioconductor pakages DESeq2 version 1.18.1[39] and IHW[40] version 1.6.0 in the framework of R software version 3.4.3 was used to perform differential gene expression analysis of cells infected with TBEV at 24versus 10 h.p.i and cells infected with TBEV and treated with TM at 8 h.p.i versus uninfected cells treated with TM at the same time point. The package is based on the negative binomial distribution (NB) to model the gene reads counts and shrinkage estimator to estimate the per-gene NB dispersion parameters. Specifically, rounded gene counts were used as input and the per-gene NB dispersion parameter was estimated using the function DESeq for DESEQ2. The RNA-seq workflow recommendations[41] were used to detect outlier data after normalization and to improve testing power, while maintaining type I error rates Independent Hypothesis Weighting was used as multiple testing procedure.

Estimated p-values for each gene were adjusted using the Benjamini–Hochberg method[42]. Genes with adjusted P < 0.05 and absolute logarithmic base twofold change >1 were selected. Data were finally analysed with the IPA software.

**Statistics**. Typically three independent experiments in triplicate repeats were conducted for each condition examined. Average values are shown with standard deviation and p-values, measured with a paired two-tailed t-test. Only significant p-values are indicated by the asterisks above the graphs (**p < 0.01 highly significant; *p < 0.05 significant). Where asterisks are missing the differences are calculated as nonsignificant (n.s).

**Reporting summary**. Further information on research design is available in the Nature Research Reporting Summary linked to this article.

## Data availability

Data underlying Figs. 1 B–J, 2 A–L, 3 A–C and E–H, 4 A–N, Supplementary Figs. 1A, B and D–F, 2 A–F, 3 A–H, and 4 A–H are provided as Source Data files. All other data are available from the corresponding author upon reasonable requests. RNAseq data have been deposited with links to BioProject accession number PRJNA474353 in the NCBI BioProject database [https://www.ncbi.nlm.nih.gov/bioproject/]. Sequence Read Archive (SRA) submission: SUB4111543—1st of June 2018, SRA accession: SRP149625.

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

## Acknowledgements

Work on flaviviruses in A.M.'s laboratory is supported by the Beneficientia Stiftung, Vaduz Lichtenstein, and by the FLAVIPOC and SEVARE projects from the Regione FVG of Italy. We thank Tatjana Avšič – Županc for the Zika virus obtained through the European Virus Archive (EVAg).

## Author contributions

T.C. and M.K.Z. performed experiments, analyzed the data, generated hypothesis, and contributed to the writing of the paper; V.F. contributed to virology experiments; L.R. contributed to the ATF6 data; Y.K. contributed to qPCR, virology, and protein analysis; D.L. performed the RNAseq and the analysis of data; A.M. conceived the work, analyzed the data, and wrote the paper.

## Additional information

**Competing interests:** The authors declare no competing interests.

