## [Peer Review File · Nature Communications]

Reviewers' comments:

Reviewer #1 (Remarks to the Author):

The report by Carletti et al (Marcello's group) provides a fascinating view of what may be a novel antiviral pathway activated by the unfolded protein response. Whilst the suggestion for the existence of such a system is indeed very interesting, the authors should provide further evidence to support this. Said additional evidence does not necessarily entail massive additional experimentation.

The authors can enhance the support of their model by the addressing the following issues:

1) The statement on page 6 'To note, at 8 h.p.i., while viral 24 yields were not yet increasing (Figure 2A), there was a significant inhibition of viral replication in the presence of TM (Figure 2B), demonstrating that this early antiviral effect is independent of any unspecific activity on the glycosylation of viral proteins.' requires further support. It is important to demonstrate that the viral proteins are not directly affected by TM and TG.

2) The data shown in 4I and 4J should have been also carried out for knockdown of PERK and ATF6.

Additionally - there is one point on the form that deserves attention. In page 9 (but also in the abstract) the use of the term 'rescue' in the sentence 'As shown in Figures 4K and L, IRF3 depletion almost completely rescued the inhibition of viral replication induced by TM treatment.' is confusing. What is rescued is viral replication not its inhibition.

Reviewer #2 (Remarks to the Author):

Manuscript by Carletti et al. Describes a functional relationship between two cellular pathways, namely the unfolded protein response (UPR) and the pattern recognition receptor (PRR) –mediated activation of innate antiviral signaling. Authors use several flaviviruses that are well known to be relevant human pathogens in the study, and they study in more detail the infection of tick-borne encephalitis virus (TBEV). By performing a detailed temporal analysis of UPR and interferon(IFN) beta induction they show that UPR activation precedes IFNbeta induction and propose that UPR-induced antiviral signaling following TBEV infection requires the IRE1 pathway. In general the manuscript is clearly written and results are well presented. Experiments are well designed and data support the conclusions claiming that there is a direct relationship between UPR and PRR-mediated activation of innate antiviral signaling in flavivirus-infected cells. These results could be useful for the advance on the complex understanding of the relationship between cellular stress pathways and innate immunity during flavivirus infection. However, some issues should be addressed to improve the study and better support the conclusions.

Specific points

1. Regarding manuscript organization, authors start the Results section with a paragraph describing a transcriptome analysis of TBEV infection that completely corresponds to Supplementary data. In my opinion this is a little confusing. Maybe authors should reconsider rearranging this section and merging it with the next piece of results because the main goal of these two sections is to show that UPR activation precedes IFNbeta induction.

2. Authors perform a detailed temporal analysis of UPR and IFNbeta activation by qPCR and Western Blotting, however in Fig. 2H they only show a time point (8h p.i.) for transcriptome analysis. This timepoint is sufficient for induction of UPR in tunicamycin (TM)-treated cells, but not to detect an enrichment in any pathway in TBEV-infected or TBEV+TM-infected cells. As they comment in the text that in TBEV-invefected cells "UPR is activated from 10 h.p.i. (Figure 1)" authors should repeat this temporal transcriptome analysis including at least 10 h post-infection, and maybe 24 h post-infecton. These additional timepoints would help the reader to interpret the data in the context of the paper because authors also show in Supplementary Fig. 1 a comparison

between 10 h p.i. vs 24 h p.i.

3. In Fig. 4 it is clearly shown that depletion of IRE1 induces significant changes in viral titers. My concern is that IRE1 was depleted using siRNA (transfection) whereas both PERK and ATF6 that did not produce any effect on viral titer were depleted using shRNAs (lentivirus). To clearly demonstrate that methodological artifacts due to the different strategies for gene silencing were not responsible for these differential results, authors should show that using identical methodologies the results are repeated.

4. Fig.2 displays that UPR activation precedes IFNbeta induction for flaviviruses, namely TBEV, DENV, WNV and ZIKV. Subsequently, authors describe the involvement of IRE1 in this phenomenon using TBEV as a model. Authors should test the role of IRE1 using other flaviviruses from the panel included in Fig. 2 to extend their results or to demonstrate that are specific for TBEV.

Reviewer #1

1) The statement on page 6 'To note, at 8 h.p.i., while viral yields were not yet increasing (Figure 2A), there was a significant inhibition of viral replication in the presence of TM (Figure 2B), demonstrating that this early antiviral effect is independent of any unspecific activity on the glycosylation of viral proteins.' requires further support. It is important to demonstrate that the viral proteins are not directly affected by TM and TG.

The effect of TM on the glycosylation of proteins is well documented, while TG acts with a different mechanism. As shown in the new Supplementary Figure 1 D&E, TM and TG both inhibit TBEV infectivity and show consequently lower levels of TBEV E protein. TM at the concentrations used in the assay induces only a partial loss of glycosylation, while TG does not affect the glycosylation of the viral E protein. Although it is possible that some of the antiviral effect of TM is due to a loss of infectivity related to the glycosylation of the E protein, there are four lines of evidence that are in favour of our working hypothesis:

- 1) The effect of TM on TBEV E glycosylation is only partial;
- 2) Treatment with TM induces also a decrease of viral RNA at early time points (8 hpi, Figure 2B) indicating a cell intrinsic antiviral effect;
- 3) Replication of TBEV, but also of other Flaviviruses, could be partially rescued by inhibiting the innate immune pathway, indicating that a significant component of the antiviral effect does not depend on a defective E protein (Figure 4 and new Supplementary Figure 4);
- 4) Treatment with TG, which does not induce changes in TBEV E protein, is also affecting viral replication and inducing an early interferon response (Figure 2E&F).

We modified accordingly the results section and added data in Supplementary Figure 1 D&E.

2) The data shown in 4I and 4J should have been also carried out for knockdown of PERK and ATF6.

According to the reviewer's suggestion we performed these experiments. As shown in the new Supplementary Figure 3 E – H, the depletion of ATF6 did not affect TBEV replication or TM-mediated rescue. Depletion of PERK by siRNA showed some degree of inhibition of viral replication, particularly in the presence of TM. However, cytotoxicity was evident in cells depleted of PERK and treated with TM, which explains the decrease of TBEV replication. To conclude, depletion of neither ATF6 nor PERK by siRNA shared the phenotype observed for IRE1 in TBEV infected cells. A comment has been added to the text and data are shown in Supplementary Figure 3 E – H.

Additionally - there is one point on the form that deserves attention. In page 9 (but also in the abstract) the use of the term 'rescue' in the sentence 'As shown in Figures 4K and L, IRF3 depletion almost completely rescued the inhibition of viral replication induced by TM treatment.' is confusing. What is rescued is viral replication not its inhibition.

As suggested by the reviewer, we have corrected this sentence within the manuscript.

Reviewer #2

1. Regarding manuscript organization, authors start the Results section with a paragraph describing a transcriptome analysis of TBEV infection that completely corresponds to Supplementary data. In my opinion this is a little confusing. Maybe authors should reconsider rearranging this section and merging it with the next piece of results because the main goal of these two sections is to show that UPR activation precedes IFNbeta induction.

The results were rearranged according to the reviewer's suggestion. The volcano plot is now shown in Figure 1 instead of Supplementary data.

2. Authors perform a detailed temporal analysis of UPR and IFNbeta activation by qPCR and Western Blotting, however in Fig. 2H they only show a time point (8h p.i.) for transcriptome analysis. This timepoint is sufficient for induction of UPR in tunicamycin (TM)-treated cells, but not to detect an enrichment in any pathway in TBEV-infected or TBEV+TM-infected cells. As they comment in the text that in TBEV-infected cells "UPR is activated from 10 h.p.i. (Figure 1)" authors should repeat this temporal transcriptome analysis including at least 10 h post-infection, and maybe 24 h post-infection. These additional timepoints would help the reader to interpret the data in the context of the paper because authors also show in Supplementary Fig. 1 a comparison between 10 h p.i. vs 24 h p.i.

UPR induction in the context of infection leads to phosphorylation of IRF3 and its nuclear translocation at early time points. Transcriptome analysis following viral infection and UPR induction showed an early signature of ISGs with antiviral activity (Figure 2H). These observations point to a direct role of the UPR in antiviral signaling. Moreover, the 8hr time point for the heatmap in Figure 3H was chosen as the time point when the IRF3 transcription factor translocate to the nucleus in conditions of infection + TM and the transcriptome analysis was designed to identify a particular set of IRF3 responsive genes that were differentially regulated under different conditions (infection vs infection + TM) at the earliest time point. In figure 3 F and G we confirm by qPCR that 2 ISGs are indeed up-regulated in response to infection + TM with an early activation at 12 hpi and increased levels at 24 hpi. As suggested by the Reviewer, to better characterize the induction of ISGs, we performed additional temporal transcription analysis of the differentially expressed ISGs: OASL, IFIH1, IFI44, IFI44L, IFIT1 and Viperin at the time

points 10 hpi and 24 hpi. As now presented in the new Supplementary Figure 2, and commented in the text, these ISGs show a pattern of early induction at 10 hpi confirming transcriptome analysis.

3. In Fig. 4 it is clearly shown that depletion of IRE1 induces significant changes in viral titers. My concern is that IRE1 was depleted using siRNA (transfection) whereas both PERK and ATF6 that did not produce any effect on viral titer were depleted using shRNAs (lentivirus). To clearly demonstrate that methodological artifacts due to the different strategies for gene silencing were not responsible for these differential results, authors should show that using identical methodologies the results are repeated.

In the original manuscript the only difference was the use of a shRNA strategy against PERK, while both ATF6 and IRE1 were depleted using a siRNA approach. To maintain the same strategy for all arms of the UPR, as suggested, we proceeded by repeating the assay for PERK using a specific siRNA. As shown in the new Supplementary Figure 3 A&B and commented in the text, depletion of PERK by siRNA confirmed the observation obtained with the shRNA against PERK. A comment has been added to the text and data are shown in Supplementary Figure 3A and 3B.

4. Fig.2 displays that UPR activation precedes IFNbeta induction for flaviviruses, namely TBEV, DENV, WNV and ZIKV. Subsequently, authors describe the involvement of IRE1 in this phenomenon using TBEV as a model. Authors should test the role of IRE1 using other flaviviruses from the panel included in Fig. 2 to extend their results or to demonstrate that are specific for TBEV.

To address this important comment we reasoned that the first step was to demonstrate that UPR priming of the IRF3-mediate innate immune response was a feature shared by flaviviruses. So we took advantage of the experimental setup where we could observe a robust rescue of TBEV replication in TM treated cells that were depleted of IRF3 (Figure 4K and 4L). Indeed, IRF3 depletion rescued WNV, ZIKV and DENV replication in TM treated cells as well (see new Supplementary Figure 4A – 4F). These experiments reinforce the notion that there is a causal link between virus-induced ER stress and innate immune sensing.

Then we investigated if the IRE1 arm of the UPR, which we could demonstrate was the critical pathway leading to cell intrinsic innate immunity following TBEV infection, shared this feature for other members of the genus. However, when we tested WNV infection we could not observe rescue of virus replication in IRE1 depleted cells (see new Supplementary Figure 4G and 4H). This observation, as well as previous data from others showing different responses in cells depleted of either IRE1, ATF6 or PERK, as already discussed in the manuscript, indicates that while TBEV appears to depend principally on the IRE1 pathway, other flaviviruses may require the contribution of the other arms of the UPR. This part will require further investigations beyond the scope of this report.

A text file with highlighted changes is attached.

Sincerely,

Alessandro Marcello, PhD
Head, Laboratory of Molecular Virology

REVIEWERS' COMMENTS:

Reviewer #1 (Remarks to the Author):

We thank the authors for their careful consideration of our previous points and for carrying out suggested experiments, now shown in two supplementary figures.

At this point and on the basis of the new data we are satisfied that the authors have provided enough evidence to strongly suggest a novel antiviral pathway activated by the unfolded protein response.

Reviewer #2 (Remarks to the Author):

The authors have adequately addressed the issues raised in my previous review.